# Energy Management in Smart Building by a Multi-Objective Optimization Model and Pascoletti-Serafini Scalarization Approach

**Zahra Foroozandeh** *(placeholder)*, **Sérgio Ramos \***, **João Soares** and **Zita Vale**

GECAD, Polytechnic of Porto, 4200-465 Porto, Portugal; zah@isep.ipp.pt (Z.F.); jan@isep.ipp.pt (J.S.); zav@isep.ipp.pt (Z.V.)
\* Correspondence: scr@isep.ipp.pt

**Abstract:** Generally, energy management in smart buildings is formulated by mixed-integer linear programming, with different optimization goals. The most targeted goals are the minimization of the electricity consumption cost, the electricity consumption value from external power grid, and peak load smoothing. All of these objectives are desirable in a smart building, however, in most of the related works, just one of these mentioned goals is considered and investigated. In this work, authors aim to consider two goals via a multi-objective framework. In this regard, a multi-objective mixed-binary linear programming is presented to minimize the total energy consumption cost and peak load in collective residential buildings, considering the scheduling of the charging/discharging process for electric vehicles and battery energy storage system. Then, the Pascoletti-Serafini scalarization approach is used to obtain the Pareto front solutions of the presented multi-objective model. In the final, the performance of the proposed model is analyzed and reported by simulating the model under two different scenarios. The results show that the total consumption cost of the residential building has been reduced 35.56% and the peak load has a 45.52% reduction.

**Keywords:** smart building; energy management; multi-objective optimization problem; mixed binary linear programming; Pascoletti-Serafini approach; Pareto front



## 1. Introduction

Currently, the use of renewable energy resources specifically solar photovoltaic (PV) panels in buildings are rapidly increasing, which contributes to supply the needed electric power of the building. Energy generation from PVs directly depends on sun radiation and weather conditions [1]. Therefore, the using of Battery Energy Storage System (BESS) and Electric Vehicles (EVs) scheduling usage, could be useful for energy management in Smart Buildings (SB) context, especially during the period of time that PV is enable to produce electricity.

So far, extensive research work has been done on the topic of building energy management in order to schedule renewable energy sources with different goals [2–6] and, as well as other studies, largely focused on the energy resource, such as the EV [7]. The model in [8] proposes to minimizes the buildings power demand and its electricity costs by optimizing the charging and discharging process of Plug-In Hybrid Electric Vehicles (PHEVs).The restrictions contain limitations for the State of Charge (SoC) of the PHEV and imposes that the electrical energy is not sold and bought at the same time for or from power grid. The main goal in the research work [9] is to minimize the daily electricity cost. Moreover, a stochastic model is presented to forecast the PV generation and building load demand. An optimal scheduling for a BESS on a Microgrid (MG) application is proposed in [10] to minimize the operating cost of the MG. The constraints of the deployed model involve the energy balance and power limitations for the both EVs and BESS. Besides, binary variables are used in order to assure that storage batteries are not charged and

discharged at the same time. In Reference [11], by considering the usage of PV generation, a Mixed-integer Linear Programming (MILP) model is proposed for EVs charging process. The main purpose of this work is to minimize the consumption energy cost for charging the EVs from power grid taking into consideration the limitation of power grid capacity. Vehicle-to-Grid (V2G) as well as a dynamic price scheme are considered as constraints. In research work presented in [12], an energy management system is proposed to minimize the peak load power demand in an SB where the contract for each apartment is assumed to be flexible. In this work, the schedule of the EVs/BESS charge and discharge processes is optimized using a Mixed Binary Linear Programming (MBLP) model in which the charging/discharging of EVs and BESS, in each time period, is modeled by binary variables. In Reference [13], a mathematical model is developed for building energy management that consisted a BESS, a PV and a fully plug-in EV. The model in this work manage the power flow between the resources by using a rule-based controller. In Reference [14], a Mixed Integer Linear Problem (MILP) model is considered in which the PV generation is included via a forecasting model, and the objective function is to minimize EVs charging cost and increase the energy consumption from the PV generation. In Reference [15], a Home Energy Management (HEM) system is considered that contains a small-scale renewable energy generation and BESS. The model is based on a MILP formulation in which V2G and demand response strategies are considered. As some recent works in this area, we can reefer to [16–18].

In this work, a Multi-Objective Mixed-Binary Linear Programming (MOMBLP) formulation is developed, which intends concomitantly to minimize the total energy consumption cost and peak load consumption. The SB considers the deployment of solar PV panels, BESS and EVs per each apartment. Moreover, it is assumed the flexibility on contract power for each apartment considering in this way a single contracted power value to supply all building facilities. In fact, the main contribution of this work is to optimize the schedule of charging and discharging process of EVs and BESS by using MOMBLP with the two mentioned objectives.

A common approach to solve numerically multi-objective optimization problems is based on scalarization approach. Scalarization methods transform a multi-objective problem into a series of parameter dependent single objective problems. The solution of each of these problems, each corresponding to a different value of the parameters, is a solution of the multi-objective problem that called Pareto point. Well-known methods based on scalarization approach are weighted-sum (WS) method [19], the $\epsilon$-constraint (EC) method [20], the normal-boundary intersection (NBI) [21–23], normal-constraint (NC) approach [22], weighted Tchebycheff (WT) method [24] and the Pascoletti-Serafini (PS) method [25,26]. In Reference [26], it is demonstrated that the PS method is more general while WS, EC, WT and NBI are special cases of the PS method. That is why authors purposed the PS method to solve MOMBLP in order to calculate the Pareto Points for two implemented objective function.

The remainder of this paper is organized as follows. In Section 2, a brief review of the multi-objective optimization problem is presented and the PS method is described. Section 3 refers the problem statement and the proposed MOMBLP model for SB. The case study and simulation results are described in Section 4. And finally, the last Section 5 provides the concluding remarks .

## 2. Multi-Objective Optimization Problem and Some Preliminaries

In this section, a brief review of the basic notations and definition of Multi-Objective Optimization Problem (MOOP) is given. Here, the following MOOP with $r$ objective function is considered.

$$MOOP \begin{cases} \min & f(x) = [f_1(x), \cdots, f_r(x)], \\ s.t. & x \in \Omega, \end{cases} \tag{1}$$

where the $\Omega = \{x \in \mathbb{R}^n \mid g(x) \leq 0, \ h(x) = 0, \ g \in \mathbb{R}^m, \ h \in \mathbb{R}^k\}$ is called *feasible set* and the image of $\Omega$ under the objective function $f(x)$ is known *Objective Space*.

When $r \geq 2$, the objective space is not naturally ordered. However, It is possible to define partial orderings. As commonly done for multi-objective problems, here the concept of Pareto optimality is used. Before introducing the Pareto optimality, some definitions are required to be identified:

**Dominance:** Two feasible points $\hat{x}, \bar{x} \in \Omega$ for the MOOP (1) are considered. It is said that the solution $\hat{x}$ *dominates* the solution $\bar{x}$, and it can be written that $\hat{x} \prec \bar{x}$, if for all $i \in \{1, \cdots, r\}$, we have $f_i(\hat{x}) \leq f_i(\bar{x})$, and there is at least one $j \in \{1, \cdots, r\}$ such that $f_j(\hat{x}) < f_j(\bar{x})$. Now, the Pareto optimality is defined by using the Dominance order.

**Pareto Minimizer(Pareto Point):** A feasible point $x^* \in \Omega$ is called a *Pareto minimizer* of MOOP (1) if there is no feasible points $x \in \Omega$ such that $x \prec x^*$. If the feasible point $x^* \in \Omega$ is a *Pareto minimizer*, then $f(x^*)$ is called a *non-dominated point*.

**Pareto Front:** The image by $f$ of the set of all *Pareto minimizers* of (1) is called the *Pareto front* (or *efficient set*).

**Ideal Point:** For each $i = 1, \cdots, r$, let $x_i^* \in \Omega$ be an optimal solution of the following single objective optimization problem. Now, the vector $a^* = [f_1(x_1^*), \cdots, f_r(x_r^*)]$ is called an *ideal point* of the MOOP (1).

$$\begin{cases} \min & f_i(x), \\ s.t. & x \in \Omega. \end{cases} \tag{2}$$

Typically, multi-objective optimization problems are solved by scalarization methods based on the reformulation of multi-objective optimization problems into a set of parameter dependent single objective optimization problems. Pareto points of the multi-objective optimization problems are generated by solving each of these single objective optimization problems corresponding to different parameters. Pascoletti-Serafini (PS) Scalarization approach is one well-known and more general of such methods [25–27]. A short overview of PS method is given in this section. In this regards, the MOOP (1) is considered. The PS scalarization method reformulates the MOOP (1) to the following single objective problem:

$$SP(a, r) \begin{cases} \min & \tau, \\ s.t. & a + \tau r - f(x) \geq 0, \\ & x \in \Omega, \tau \in \mathbb{R}, \end{cases} \tag{3}$$

with parameters $a, r \in \mathbb{R}^r$. To simplicity,it is assumed that $r = 2$ and we apply the method for bi-criteria problems. The Pareto front of MOOP (1) is obtained by solving the associated $SP(a, r)$ problem for different parameters $(a, r)$ that the parameter $a$ is considered on the line $a \in \{y \mid y = \epsilon x_1^* + (1 - \epsilon)x_2^*, \ \epsilon \in [0, 1]\}$. Note that $x_1^*$ and $x_2^*$ are the optimal solutions of problem (2) with $r = 2$. In Reference [26], the sufficient and necessary conditions for optimal Pareto points of MOOP are given and also show that all Pareto points derived from MOOP can be obtained by PS scalarization.

## 3. Problem Description and Mathematical Model

In this work, a residential building is considered which contains Photo Voltaic (PV) generation panels, Electric Vehicles (EVs) and a Battery Energy Storage System (BESS). The main purpose in this paper is to find out the best charging and discharging scheduling process for both EVs and BESS, in order to minimize the energy consumption cost from power grid as well as smoothing peek load consumption in the considering time-period. To achieve these proposal goals some assumption in SB are previously considered:

- Solar photovoltaic production is considered for self-consumption but with the possibility of selling the surplus to the power grid;
- It is considered that each EV leaves and arrives to the building once a day. All EVs are connected to the electric network as soon as they get home.

- The arrival and departure times are known and for each EV the initial SoC is known at the arrival time. The EV battery can be charged/discharged between arrival and departure.

Now, the considering problem is formulated by a MOOP with two objective functions ($r = 2$). In order to describe the model, the used parameters and decision variables are summarized in Tables 1 and 2.

Note that in Table 2, the binary variables $\alpha_{\text{EV}}(i, j)$ and $\beta_{\text{EV}}(i, j)$ are used to define the charging and discharging state of $j$-th EV in $i$-th time-slot. $\alpha_{\text{EV}}(i, j) = 1$ ($\beta_{\text{EV}}(i, j) = 1$) means that the battery of $j$-th EV is charging (discharging) in time-slot $i$. The binary variables $\alpha_{\text{BE}}(i)$ and $\beta_{\text{BE}}(i)$ are similarly used for charging/discharging state of BESS. Moreover, if $j$-th EV is out of SB in time-slot $i$, then the variable $S_{\text{EV}}(i, j)$ is meaningless and should not be considered in the model. On the other hand, the index $i$ of $S_{\text{EV}}(i, j)$ is considered in $\{1, \ldots, I\}$. Indeed, for simplicity in presentation, authors considered index $i \in \{1, \ldots, I\}$ for $S_{\text{EV}}$ and and this will be taken into account when formulating the objective function and restrictions.

**Table 1.** Parameters of the model.

| Parameter | Index | Description |
|---|---|---|
| $I$ | | Number of time-slots per Time-Study |
| $\tau$ | | Time-slot duration (hour) |
| $J$ | | Number of apartments (or EVs) in the building |
| $T_{\text{EV}}^{\text{in}}(j)$ | $j \in \{1, \ldots, J\}$, | The number of period-time in which $j$-th EV enters to the parking |
| $T_{\text{EV}}^{\text{out}}(j)$ | $j \in \{1, \ldots, J\}$, | The number of period-time in which $j$-th EV leaves the parking |
| $S_{\text{EV}}^{\max}(j)$ | $j \in \{1, \ldots, J\}$ | Maximum allowable State of Charge(SoC) of $j$-th EV |
| $S_{\text{EV}}^{\text{initial}}(j)$ | $j \in \{1, \ldots, J\}$, | The initial SoC of $j$-th EV at the beginning departure in $T_{\text{EV}}^{\text{in}}(j)$ |
| $S_{\text{EV}}^{\text{min\_out}}(j)$ | $j \in \{1, \ldots, J\}$, | The minimum allowable SoC for $j$-th EV at exit time |
| $S_{\text{BE}}^{\max}$ | | Maximum SoC for BESS |
| $S_{\text{BE}}^{\text{initial}}$ | | Initial SoC for BESS at the beginning of time-period |
| $P_{\text{SB}}(i)$ | $i \in \{1, \ldots, I\}$ | Total power demand of SB at period $i$ |
| $P_{\text{PV}}(i)$ | $i \in \{1, \ldots, I\}$ | Total generated power by PV at period $i$ |
| $P_{\text{G}}^{\max}(i)$ | $i \in \{1, \ldots, I\}$ | Maximum power that can got from Grid at time-slot $i$ |
| $C_{\text{G}}^{\text{buy}}(i)$ | $i \in \{1, \ldots, I\}$ | Purchased electricity cost from grid in $i$-th time-slot |
| $C_{\text{G}}^{\text{sell}}(i)$ | $i \in \{1, \ldots, I\}$ | Sell electricity cost to grid in $i$-th time-slot |
| $P_{\text{EV}}^{\text{ch}}(j)$ | $j \in \{1, \ldots, J\}$ | Active power related to the charging process of the $j$-th EV |
| $P_{\text{EV}}^{\text{diss}}(j)$ | $j \in \{1, \ldots, J\}$ | Active power related to the discharging process of the $j$-th EV |
| $E_{\text{EV}}^{\text{ch}}(j)$ | $j \in \{1, \ldots, J\}$ | The charge efficiency of$j$-th EV |
| $E_{\text{EV}}^{\text{diss}}(j)$ | $j \in \{1, \ldots, J\}$ | The discharge efficiency of $j$-th EV |
| $P_{\text{BE}}^{\text{ch}}(i)$ | $i \in \{1, \ldots, I\}$ | Active power of the charging process of the BESS in period $i$ |
| $P_{\text{BE}}^{\text{diss}}(i)$ | $i \in \{1, \ldots, I\}$ | Active power of the discharging process of BESS in period $i$ |

<div align="center">

**Table 2.** Decision variables of the model.

</div>

| Variable | Type | Index | Description |
|---|---|---|---|
| $\alpha_{\text{EV}}(i,j)$ | $\{0,1\}$ | $i \in \{1,\dots,I\}, j \in \{1,\dots,J\}$ | $j$-th EV charging process in period $i$ |
| $\beta_{\text{EV}}(i,j)$ | $\{0,1\}$ | $i \in \{1,\dots,I\}, j \in \{1,\dots,J\}$ | $j$-th EV discharging process in period $i$ |
| $\alpha_{\text{BE}}(i)$ | $\{0,1\}$ | $i \in \{1,\dots,I\}$ | BESS charging process in period $i$ |
| $\beta_{\text{BE}}(i)$ | $\{0,1\}$ | $i \in \{1,\dots,I\}$ | BESS discharging process in period $i$ |
| $S_{\text{EV}}(i,j)$ | $\mathbb{R}_0^+$ | $i \in \{1,\dots,I\}, j \in \{1,\dots,J\}$ | SoC of the $j$-th EV at the start of period $i$ |
| $S_{\text{BE}}(i)$ | $\mathbb{R}_0^+$ | $i \in \{1,\dots,I\}$ | SoC of the BESS at the start of period $i$ |
| $P_{\text{G}}(i)$ | $\mathbb{R}_0^+$ | $i \in \{1,\dots,I\}$ | Active power extracted from the grid in period $i$ |
| $P_{\text{G}\rightarrow\text{BE}}(i)$ | $\mathbb{R}_0^+$ | $i \in \{1,\dots,I\}$ | Active power of charging the BESS by grid in period $i$ |
| $P_{\text{G}\rightarrow\text{EV}}(i,j)$ | $\mathbb{R}_0^+$ | $i \in \{1,\dots,I\}, j \in \{1,\dots,J\}$ | Active power of charging the $j$-th EV by grid in period $i$ |
| $P_{\text{EV}\rightarrow\text{B}}(i,j)$ | $\mathbb{R}_0^+$ | $i \in \{1,\dots,I\}, j \in \{1,\dots,J\}$ | Active power of discharging of $j$-th EV to SB in period $i$. |
| $P_{\text{PV}\rightarrow\text{B}}(i)$ | $\mathbb{R}_0^+$ | $i \in \{1,\dots,I\}$ | Active power from PV to SB in period $i$ |
| $P_{\text{PV}\rightarrow\text{BE}}(i)$ | $\mathbb{R}_0^+$ | $i \in \{1,\dots,I\}$ | Active power from PV to BESS in period $i$ |
| $P_{\text{PV}\rightarrow\text{G}}(i)$ | $\mathbb{R}_0^+$ | $i \in \{1,\dots,I\}$ | Active power from PV to grid in period $i$ |
| $P_{\text{BE}\rightarrow\text{G}}(i)$ | $\mathbb{R}_0^+.$ | $i \in \{1,\dots,I\}$ | Active power from BESS to grid in period $i$ |

*Formulation of MOOP*

In this section, a MOOP model is formulated for residential building considering two objective functions. The first objective function $f_1$ intends to minimize the total consumption energy cost as

$$f_1 = \min\Big\{ \sum_{i=1}^{I} \Big( P_{\text{G}\rightarrow\text{B}}(i) + P_{\text{G}\rightarrow\text{BE}}(i) + \sum_{j=1}^{J} P_{\text{G}\rightarrow\text{EV}}(i,j) \Big) C_{\text{G}}^{\text{buy}} - \tag{4}$$

$$\sum_{i=1}^{I} \Big( P_{\text{PV}\rightarrow\text{G}} + P_{\text{BE}\rightarrow\text{G}} + \sum_{j=1}^{J} P_{\text{EV}\rightarrow\text{G}}(i,j) \Big) C_{\text{G}}^{\text{sell}} \Big\}.$$

In (4), the first summation represents the total amount of electrical energy bought from power grid used by SBs, BESS and EVs. The second summation in (4) corresponds the total amount of electrical energy that it is injected by PVs, BESS and EVs to the external power grid.

And the second objective function $f_2$, represented in (5), minimizes the peak load consumption of the SB. In this regards, the following objective function was defined:

$$f_2 = \min\Big\{ \max_{i\in\mathbb{I}} P_{\text{G}}(i). \Big\} \tag{5}$$

By defining the slack variable $z$, the objective function (5) is reformulated as

$$\begin{aligned} f_2 &= \min \ z, \\ &\quad P_{\text{G}}(i) \leq z \\ &\quad z \geq 0. \end{aligned} \tag{6}$$

In general, the proposed MOOP model for SBs is represented by Equations (7a)–(7p).

$$\text{Minimize} \quad \mathcal{J} = [f_1, f_2], \tag{7a}$$

$$\text{S.t.} \quad S_{\text{EV}}(i+1,j) = S_{\text{EV}}(i,j) + \left[ P_{\text{G}\to\text{EV}}(i,j)E_{\text{EV}}^{\text{ch}} - (P_{\text{EV}\to\text{G}}(i,j) + P_{\text{EV}\to\text{B}}(i,j))/E_{\text{EV}}^{\text{diss}} \right], i \in \{1,\ldots,I-1\} \tag{7b}$$

$$S_{\text{EV}}(1,j) = S_{\text{EV}}^{\text{initial}}(j), \quad j \in \{1,\ldots,J\}, \tag{7c}$$

$$P_{\text{G}\to\text{EV}}(i,j) \leq \alpha_{\text{EV}}(i,j)P_{\text{EV}}^{\text{ch}}(j)\tau, \quad i \in \{1,\ldots,I\}, j \in \{1,\ldots,J\} \tag{7d}$$

$$P_{\text{EV}\to\text{G}}(i,j) + P_{\text{EV}\to\text{B}}(i,j) \leq \beta_{\text{EV}}(i,j)P_{\text{EV}}^{\text{diss}}(j)\tau, \quad i \in \{1,\ldots,I\}, j \in \{1,\ldots,J\} \tag{7e}$$

$$0 \leq S_{\text{EV}}(i,j) \leq S_{\text{EV}}^{\text{max}}(j), \quad i \in \{1,\ldots,I\}, j \in \{1,\ldots,J\}, \tag{7f}$$

$$S_{\text{EV}}(T_{\text{EV}}^{\text{out}}(j)-1,j) \geq S_{\text{EV}}^{\text{min\_out}}(j), \quad j \in \{1,\ldots,J\}, \tag{7g}$$

$$S_{\text{EV}}(i,j) = 0, \quad j \in \{1,\ldots,J\}, \ i \in \{1,\ldots,T_{\text{EV}}^{\text{in}}(j)-1\} \cup \{T_{\text{EV}}^{\text{out}}(j)+1,\ldots,I\}, \tag{7h}$$

$$\alpha_{\text{EV}}(i,j) + \beta_{\text{EV}}(i,j) \leq 1, \ i \in \{1,\ldots,I\}, j \in \{1,\ldots,J\}, \tag{7i}$$

$$S_{\text{BE}}(i+1) = S_{\text{BE}}(i) + \left[ (P_{\text{G}\to\text{BE}}(i) + P_{\text{PV}\to\text{BE}}(i))E_{\text{BE}}^{\text{ch}} - (P_{\text{BE}\to\text{G}}(i) + P_{\text{BE}\to\text{B}}(i))/E_{\text{BE}}^{\text{diss}} \right], \ i \in \{1,\ldots,I\}, \tag{7j}$$

$$S_{\text{BE}}^{\text{min}} \leq S_{\text{BE}}(i) \leq S_{\text{BE}}^{\text{max}}, \quad i \in \{1,\ldots,I\}, \tag{7k}$$

$$P_{\text{G}\to\text{BE}}(i) + P_{\text{PV}\to\text{BE}}(i) \leq \alpha_{\text{BE}}(i).P_{\text{BE}}^{\text{ch}}\tau, \quad i \in \{1,\ldots,I\}, \tag{7l}$$

$$P_{\text{BE}\to\text{G}}(i) + P_{\text{BE}\to\text{B}}(i) \leq \beta_{\text{BE}}(i)P_{\text{BE}}^{\text{diss}}\tau, \quad i \in \{1,\ldots,I\}, \tag{7m}$$

$$\alpha_{\text{BE}}(i) + \beta_{\text{BE}}(i) \leq 1, \quad i \in \{1,\ldots,I\}, \tag{7n}$$

$$0 \leq P_{\text{G}}(i) \leq P_{\text{G}}^{\text{max}}, \quad i \in \{1,\ldots,I\}, \tag{7o}$$

$$P_{\text{G}}(i) \leq z, \quad z \geq 0. \tag{7p}$$

We recall that the decision variables of the above model is presented in Table 2. According to this table, we find that for $I$ days and $J$ apartment, the number of decision variables is $(9+5J)I+1$. In the above model, Equation (7b–i) represent the EVs constraints in which (7b) shows the SoC updating of EVs in each time slots due to EVs charging and discharging process, knowing initial charge state condition (7c). The Equation (7d) describes the maximum charging value for EVs. Note that, the binary variable $\alpha_{\text{EV}}(i,j)$ in this equation shows the charging state of the $j$-th EV in the $i$-th time-slot. In addition, the maximum EVs discharging value is presented in (7e) in which discharging state of $j$-th EV in $i$-th time-slot is presented by binary variable $\beta_{\text{EV}}(i,j)$. The capacity of each EVs during the time period and in the last time slot are defined by Constraints (7f) and (7g) respectively. In time slot $i \in \{1,\ldots,T_{\text{EV}}^{\text{in}}(j)-1\} \cup \{T_{\text{EV}}^{\text{out}}(j)+1,\ldots,I\}$, the $j$-th EV is outside of the building and the charging and discharging process does not occurs. In this regards, the Equation (7h) is considered in this time-slot. And finally the Constraints (7i) ensure that the charging and discharging of EVs do not occur at the same time. Note that, If any tenant ($j$) does not have an Electric Vehicle, it is possible to consider it in the model by $S_{\text{EV}}^{\text{max}}(j) = 0$.

In similar process, the Equations (7j)–(7n) represent the BESS constraints in which the updating SoC process of BESS is presented by Equation (7j) with known initial value and the equations (7k) limits the capacity of BESS. The Equations (7l) and (7m) describe the maximum value of charge/discharging process of BESS in which the binary variables $\alpha_{\text{BE}}(i)$ and $\beta_{\text{BE}}(i)$ present the charge/discharging state of BESS in time slot $i$. Moreover, the bound constraints for consuming energy from external power network is considered by (7o).

## 4. Case Study

In this section, a residential building containing 15 apartments and three PV sources generation with a capacity 3.68 kW is considered as a case study that each apartment has one Electrical Vehicle. The value of the mentioned parameters in the model (7), such as power demanded value from apartments $P_{\text{SB}}$, the PVs generated power $P_{\text{PV}}$ and arrival/departure time of EVs $T_{\text{EV}}^{\text{in}}/T_{\text{EV}}^{\text{out}}$ are recorded for each 15 minutes.

Real data bases concerning load consumption and PV generation were used. There are always problems with databases, which is required a data pre-processing phase in order

to "clean" the database. It was identified that some recorded data were missed. In order to fill in the lack of data values, and in this way, to enable the use of database for study, a regression model and an adjacent interpolation approaches were implemented.

In the proposed optimization model (7a)–(7p), any time-period (in days) can be selected. However, to sake of simplicity in the numerical simulation, the results of the model for one day and $\tau = 0.15$ minutes, consequently, the time-slot has $I = 24 \times 4 = 96$ time-slots.

Moreover, as already mentioned, it was assumed that the residential building is equipped by an BESS and each apartment has one EV with the following characteristics

$$S_{EV}^{max} = 27.2 \text{ kW h}, \quad P_{EV}^{ch} = 3.7 \text{ kW}, \quad P_{EV}^{diss} = 3.33 \text{ kW} \tag{8}$$

$$S_{BE}^{max} = 50 \text{ kW h}, \quad P_{BE}^{ch} = 6.3 \text{ kW}, \quad P_{BE}^{diss} = 5.67 \text{ kW}. \tag{9}$$

In addition, the initial State of Charge (SoC) is considered $S_{BE}^{initial} = 0$ and the initial SoC of EVs at the arrival time $S_{EV}^{initial}(j)$ are set randomly .

*Simulation Results*

Before applying the PS approach for (7), it is required to find out the ideal point $a$. In this sense, the CPLEX solver was used in order to solve the single objective optimization problem (2). The limiting weights $\epsilon = 0$ and $\epsilon = 1$ are considered to characterize the optimal solution of single-objective problems in relation to the problem (7) as shown in (10) and (11):

$$\epsilon = 0, \quad [f_1(x_1^*), f_2(x_1^*)] = [39.031\,7 \text{ EUR}, 8.599\,0 \text{ kW h}], \tag{10}$$

$$\epsilon = 1, \quad [f_1(x_2^*), f_2(x_2^*)] = [54.891\,9 \text{ EUR}, 4.703\,3 \text{ kW h}]. \tag{11}$$

Note that, when $\epsilon = 0$, the single-Objective problem is solved in order to minimize the total energy cost(the optimal achieved value is 39.031 7 EUR). when $\epsilon = 1$, the single-Objective problem is solved in order to minimize the demanded peak load (the optimal achieved value is 4.703 3 kW h).

So, the idea point for MOOP problem (7) is to obtain $a = [f_1(x_1^*), f_2(x_2^*)] = [39.031\,7 \text{ EUR}, 4.703\,3 \text{ kW h}]$.

Now, the PS approach in Section 2 is applied for problem (7) with direction $r = [1, 1]$. The approximation of Pareto front is reported in Figure 1. For better vision, the objective functions $f_1$ and $f_2$, as a function of $\epsilon \in [0, 1]$, are plotted in Figure 2.

The Figure 1 shows the variation of the both objective function according to the variation of parameter $\epsilon \in [0, 1]$ in the PS method. And, Figure 2a depicts the distance between each point of the Figure 1 (Pareto points) with the ideal point (identified in Figure 1) in order to find the best point. Figure 2b,c show the changes of each objective function relative to the changes of parameter $\epsilon \in [0, 1]$ in the PS method.

In order to find the most appropriate Pareto point, the overall of Pareto front is optimized by using the objective function $\mathcal{K} = ||(f_1, f_2) - a||_2$ , that is, measuring the distance of each Pareto point with the ideal point. Function $\mathcal{K}$ as a function of $\epsilon \in [0, 1]$ is reported in the left side of Figure 2. It can be seen that the objective function $\mathcal{K}$ reaches its minimum value at $\mathcal{K} = 0.292\,2$ that is attained at $\epsilon = 0.834$ with following results

$$\epsilon = 0.834, \quad f_1 = 39.2349 \text{ EUR}, \quad f_2 = 4.9134 \text{ kW h}, \quad \text{Computation Time} = 1.1015 \text{ s}. \tag{12}$$

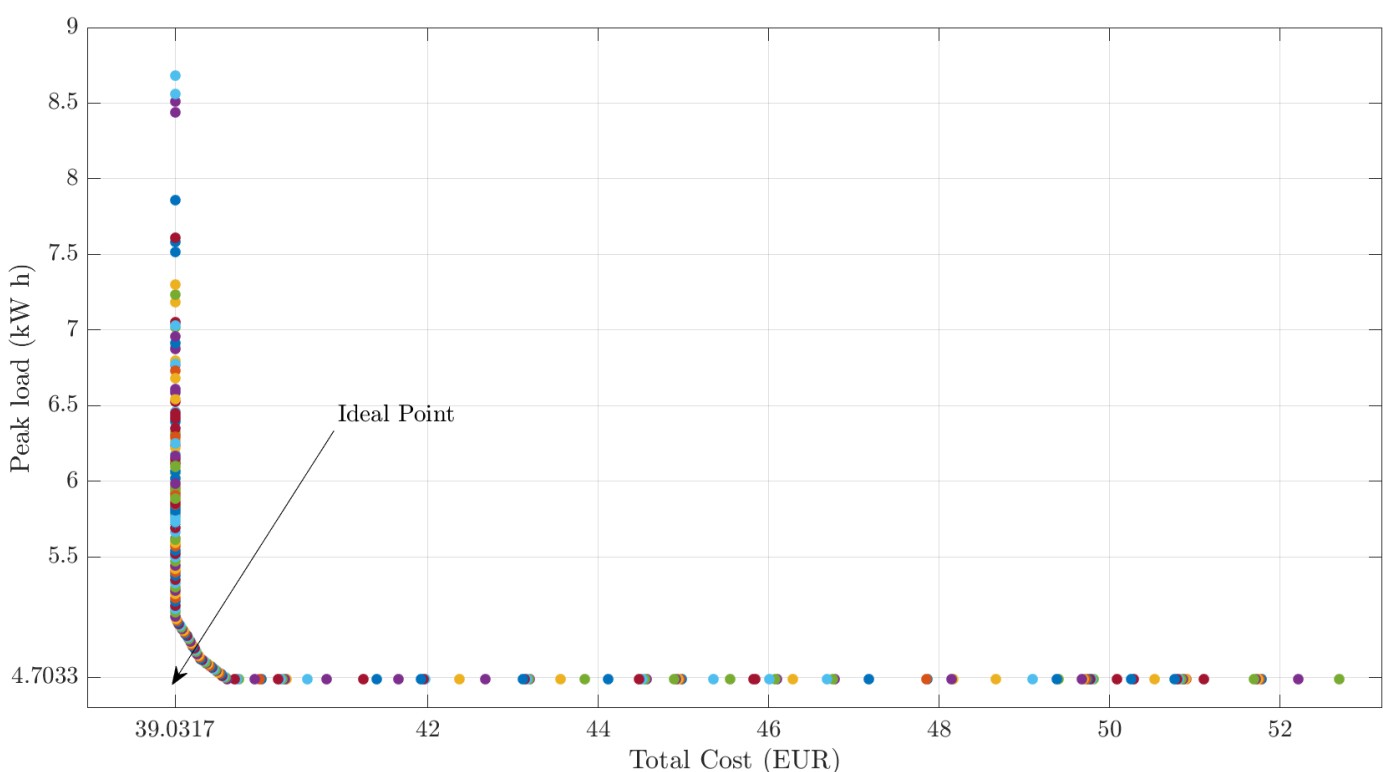

**Figure 1.** The obtained Pareto front for problem (7).

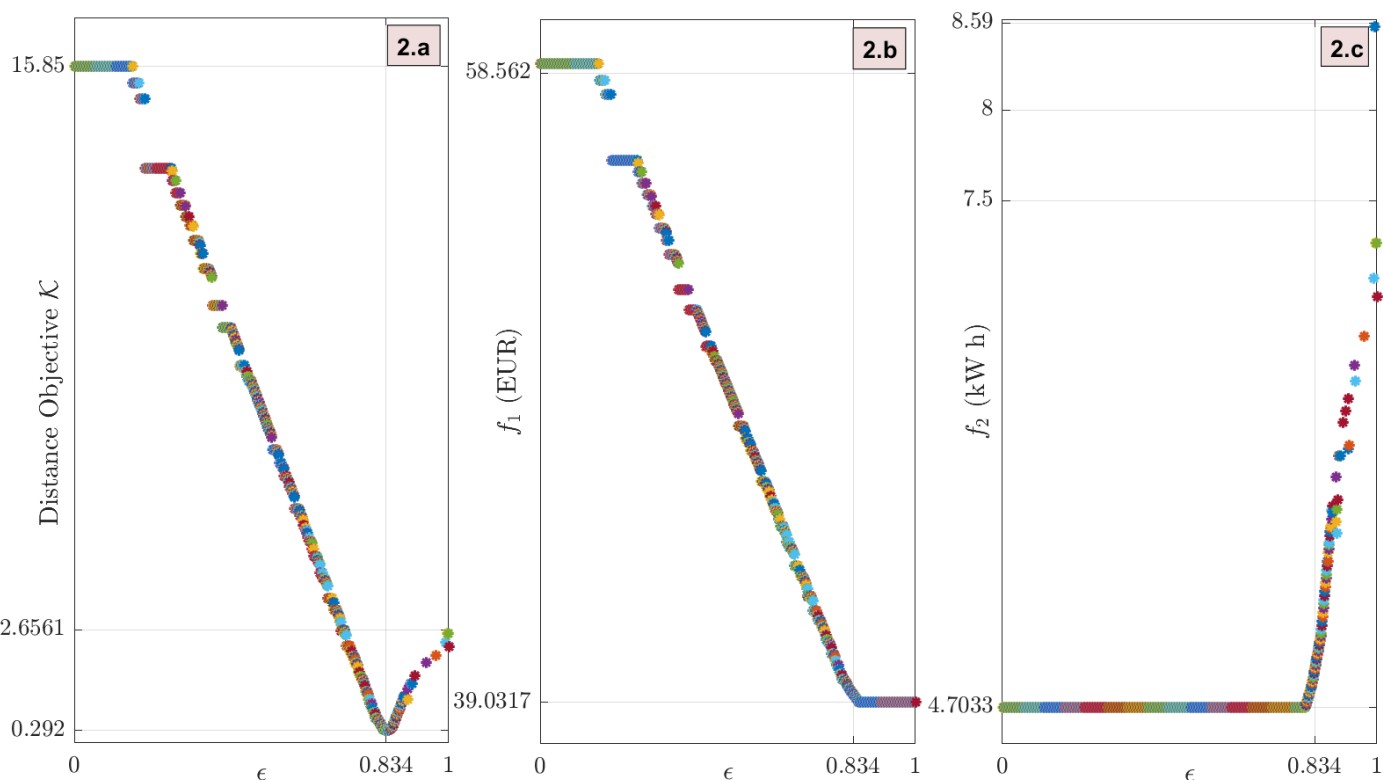

**Figure 2.** (**a**) Distance of Pareto Points from the Ideal Point, (**b**) Objective Function $f_1$ respect to $\epsilon$, (**c**) Objective Function $f_2$ respect to $\epsilon$.

The electrical energy consumption from external power grid, the generated power by PVs, the building power demand and the used energy for charging/discharging EVs,

corresponding to $\epsilon = 0.834$ are plotted in Figure 3. Moreover, the interactions among energy resources are specified by different colors.

In Figure 3, it is possible to see the obtained results for one day. Figure 3a represents the electrical energy that is supplied from external power grid. Figure 3b represents the generation energy from PV panels. Figure 3c depicts the consumption load profile concerning the residential building. Figure 3d represents the obtained EV's charging and discharging process, taking into consideration the use of the BESS, PV and external power grid. Figure 3e shows the obtained BESS charging and discharging process results.

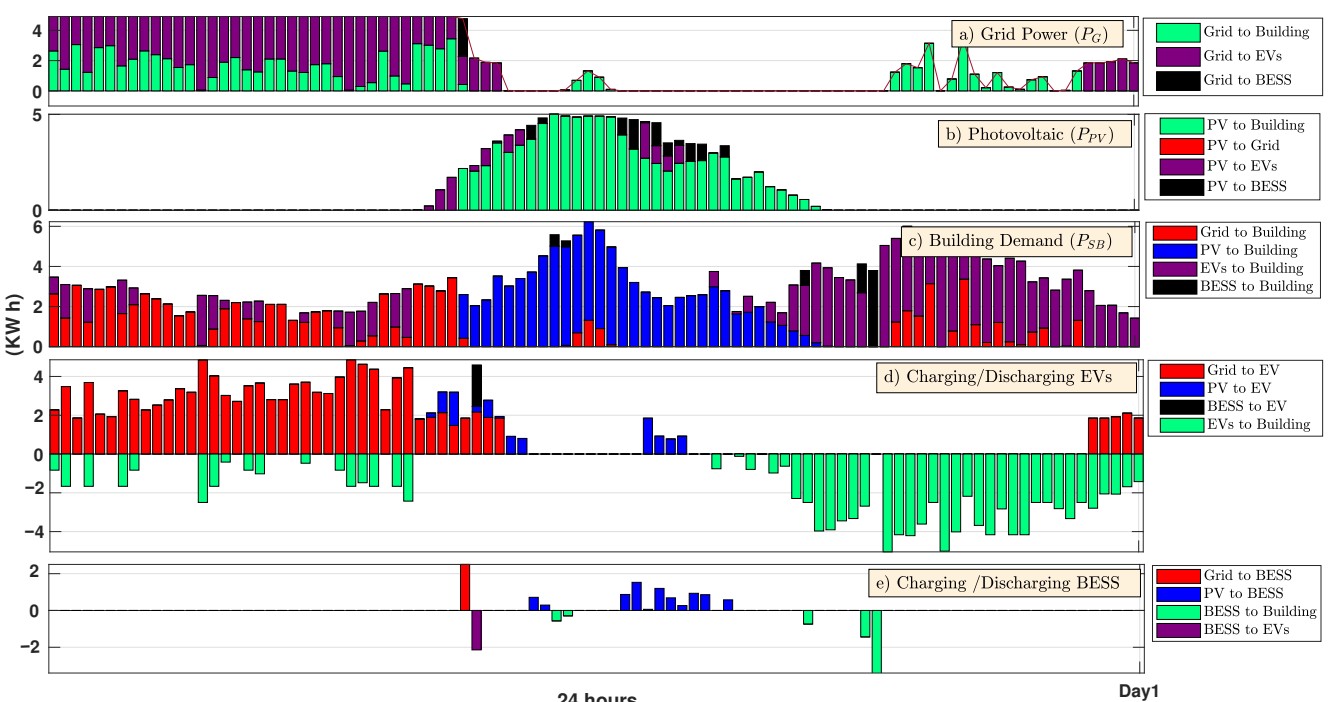

**Figure 3.** Trace of power among Grid, Building's apartments, Photo Voltaics (PVs), Electric Vehicles (EVs) and Battery Energy Storage System (BESS) corresponding to $\epsilon = 0.834$.

As shown in Figure 3, in some step-times, some PV generated power is used for charging EVs' and BESS. the EVs and BESS are discharged in order to reduce the consumed power from external power grid and at the other time-slots. Moreover, all the PV generated power used by smart building, that is for charging BESS (if it is available), to charge EVs or to supply the electrical consumers. During the off-peak hours, costumers are supplied mainly by external power grid, EVs and also a little slight contribution from BESS. Mainly, during the off-peak hours, EVs are charged from external power grid and they are also charge by BESS and PV during Half-peak hours. The contribution from EV to supply the building occurs at off-peak hours. The BESS takes advantage from the PV generation source and contribute to charge EVs and to supply some energy building demand.

Finally, in order to show the efficiency of the proposed model, the obtained results (total consumption costs and the maximum value of the demanded load) were compared with the reference case study that neither considered the flexibility of contracted power for each apartment nor the EVs discharging process [12]. The compared results are reported in Table 3.

**Table 3.** Comparison of Proposed Model (MOMBLP) with Reference Case Study for 1 day from external Supplier.

|  | **Total Cost (EUR)** | **Peak Load (kW h)** |
|---|---|---|
| MOMBLP Model | 39.2349 | 4.9134 |
| Reference Case Study [12] | 60.8838 | 9.0190 |

According to the obtained results, shown in Table 3, it is possible to verify a significant reduction in both objective functions. Moreover, the total consumption cost of the residential building has been reduced 35.56% and the peak load has reduced 45.52% in comparison with the reference case study [12].

## 5. Conclusions

Traditionally, each residential consumers has its own electrical contract power. This approach proposes the flexibility of the customer contracted power taking into consideration the energy management resources, namely, photovoltaic generation, battery energy storage system and electrical vehicles usage. A multi-objective mixed binary linear problem (MOMBLP) is suggested to model the proposed idea in order to minimize building peak load consumption and also aiming to reduce the total cost of the building from external consumption power grid. Note that, The binary variables in the proposed MOMBLP are used to find automatically the scheduling of the charging and discharging process of the BESS and EVs. Their proposed approach considered that apartments have flexibility of the contract power that is, the building has just one contract power. In order to validate the model, a real residential building was considered that contains 15 apartments, solar PVs generation panels, EVs from each apartment and a BESS. The PS scalarization method was used to solve the proposed Multi objective optimization problem since then it is a general method for solving multi-objective problems. The obtained Pareto front and the objective functions respected to $\epsilon$ is reported. A new objective function is defined to search over the Pareto front and discover the appropriate Pareto point. Finally, the building energy resources are obtained showing the contribution of each one.

**Author Contributions:** Conceptualization, Z.F., S.R. and J.S.; methodology, Z.F., J.S. and S.R.; data curation, J.S.; formal analysis, Z.F. and S.R.; funding acquisition, S.R. and J.S.; investigation, Z.F.; validation, Z.V., S.R.; supervision, Z.V. All authors have read and agreed to the published version of the manuscript.

**Funding:** This work has received funding from FEDER Funds through COMPETE program and from National Funds through FCT under the project BENEFICE–PTDC/EEI-EEE/29070/2017 and UIDB/00760/2020 under CEECIND/02814/2017 grant.

**Institutional Review Board Statement:** Not applicable.

**Informed Consent Statement:** Not applicable.

**Data Availability Statement:** Data are available from the authors.

**Conflicts of Interest:** The authors declare no conflict of interest.

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
