# Peer review of "Energy Management in Smart Building by a Multi-Objective Optimization Model and Pascoletti-Serafini Scalarization Approach"

_processes, doi:10.3390/pr9020257_

Round 1

Reviewer 1 Report

Paper "Energy Management in Smart Building by A Multi-Objective OptimizationModel and Pascoletti-Serafini Scalarization Approach" raises an important and interesting topic but there are issues which should be improved.

    1. The literature review should be extended at least 10 positions. This topic is up to date and a lot of paper were published in recent years.
    2. I didn't see if optimization was done for one day or for a longer time period. According to me variability of PV generation in different year seasons should be included. For exaple generation in summer in much higher than in winter. 
    3. I didn't see the residential profile which was used for calculations. Please explain which sape of profile did You use.
    4. Conclusions should be extended and contain explanation why Your model is better than others one.
    5. Were the capital costs included in the optimization model?
    6. Which was the computational time?
    7. How many variables were during optimization?
    8. Please describe better figures 1 and 2.
    9. I didn't see what is the capacity of PV and energy storege?
    10. Please explain why You use this type of optimization.

Author Response

The authors would like to express their gratitude to the reviewer 1 for his valuable comments that give us the opportunity to improve the paper. The authors present their response below.

  1. The literature review should be extended at least 10 positions. This topic is up to date and a lot of paper were published in recent years.

Change: In accordance with this comment, 9 references were added to the new version of the paper. Please see Refs. [1], [7], [9], [13-18].

  1. I didn't see if optimization was done for one day or for a longer time period. According to me variability of PV generation in different year seasons should be included. For example generation in summer in much higher than in winter.

Reply: Thank you for your comment. In the proposed optimization model, any time-period (in days) can be selected. However, to sake of brevity in the numerical simulation, the results of the model for one day is reported. Obviously, any day of year can be considered. 

Change: Based on this comment, to clarify the mentioned point, a paragraph in Section 4 is added. Please see Lines 144 to 146.

  1. I didn't see the residential profile which was used for calculations. Please explain which shape of profile you used.

Reply: Thank you for this comment. In Section 4, the residential profile is briefly reported. However, the details are massive and not reported in the case study. However, in Figure 3, it is possible to see the residential profile for one day. Figure 3.a) represents the electrical energy that is supplied from external power grid. Figure 3.b) represents the generation energy from PV panels. Figure 3.c) depicts the consumption load profile concerning the residential building. Figure 3.d) represents the obtained EV’s charging and discharging process, taking into consideration the use of the BESS, PV and external power grid. Figure 3.e) shows the obtained BESS charging and discharging process.

Change: Authors have revised this point in the new manuscript. A new version of Figure 3 is considered as well as the identification of the used consumption load profile.

Please see in Section 4 – Case Study: from line 167 to 171.

  1. Conclusions should be extended and contain explanation why Your model is better than others one.

Reply: Thank you for this question. Authors revised the conclusion section. Also, to show the impact of our proposed method, Table 3 was added in which the total cost and peak load of the proposed method are comprised with a previous case study, also developed by authors. Moreover, it was considered a multi-objective model, where it provides suitable information for decision makers.

Change: The conclusions section was revised and a comparison made in Table 3, Chapter 4.

Please see in Section 4 – Case Study: from line 181 to 187.

  1. Were the capital costs included in the optimization model?

Reply: The capital costs are not considered in our model. This need more work and can be considered in another work.

  1. Which was the computational time?

Change: Thank you for your suggestion. The consuming computational time for computing the solution is added in Eq. (12).

  1. How many variables were during optimization?

Change: The number of decision variables in the proposed model is specified in the new manuscript.

Please see lines 117-118.

  1. Please describe better figures 1 and 2.

 Change: Following the reviewer suggestion, Authors have described Figures 1 and 2

Please see Section 4 – Case Study: frim lines 160 to 163.

  1. I didn't see what is the capacity of PV and energy storage?

Change: We are sorry for this issue. In the revised manuscript, the capacity of PVs is more clearly reported. Please see in Chapter 4, Lines: 136-137. Also the energy storage of EVs and BESS are mentioned in Eq. (8) and (9).

  1. Please explain why You use this type of optimization.

 Reply: The proposed approach considered multi-objective mixed binary linear problem because: 1) the binary variables are used to find automatically the scheduling of the charging and discharging process of the BESS and EVs. 2) Multi-objective optimization paradigm is used to minimize two objective function at same time (minimize consumption peak loads and total costs). This type of optimization is desirable for decision makers to select suitable solutions. A Pascoletti-Serafini Scalarization Approach is proposed since then it is a general method for solving multi-objective problems.

Change: The mentioned points are highlighted in the Conclusion of the revised paper.

Please see Chapter 5: Conclusion, Lines: 189-197 and lines: 199-200.

Reviewer 2 Report

Optimizing the cost of electricity consumption and peak loads in residential buildings by charging and discharging the Electric Vehicles (EVs) and Battery Energy Storage System (BESS) is an interesting topic. However, it has been demonstrated in several studies that energy storage is an adequate solution for the energy obtained by the photovoltaic modules to fit the energy demand profile of the users. So, it is necessary to emphasize the novelty of the research with respect to the existing literature and to extend the state of the art due to the fact that there are numerous scientific studies of interest related to the subject of the research.

The research aims to minimize the total cost of energy consumption and peak load in collective residential buildings, but it does not show the percentage of energy savings or peak load reduction obtained from implementing the model proposed in the case under analysis.

It would be appropriate to compare the results obtained in the research with the results of other studies of the state of the art, in order to be able to evaluate the potential of the proposed model with respect to other solutions and to highlight its advantages and limitations. For example, estimating the arrival time of electric vehicles is difficult, as atmospheric conditions can influence how often a vehicle is used on the road. In addition, the model considers that all users have an electric vehicle, which is a difficult limitation to achieve at present. Besides, the abstract does not present numerical results obtained during the research.

On the one hand, in this type of study it is convenient to analyze the profitability of the proposed installation compared to other solutions already carried out. On the other hand, in the proposed case, the characteristics of the photovoltaic installation and the solar radiation values are not presented.

Author Response

The authors would like to express their gratitude to the reviewer 2 for his valuable comments that give us the opportunity to improve the paper. The authors present their response below.

  1. Optimizing the cost of electricity consumption and peak loads in residential buildings by charging and discharging the Electric Vehicles (EVs) and Battery Energy Storage System (BESS) is an interesting topic. However, it has been demonstrated in several studies that energy storage is an adequate solution for the energy obtained by the photovoltaic modules to fit the energy demand profile of the users. So, it is necessary to emphasize the novelty of the research with respect to the existing literature and to extend the state of the art due to the fact that there are numerous scientific studies of interest related to the subject of the research.

Reply: Thank you for your suggestion. As you mentioned, the referred ideas are separately considered in several studies. However, in this paper, the electricity consumption cost and building contracted power are considered simultaneously. Moreover, the proposed mathematical model is more flexible with time of study and is simply stated in comparison with the last relevant works. However, the main contribution of this paper is to consider one contract power for whole building, instead of individual contracted power per each tenant, and scheduling the charge/discharge process by using a multi-objective optimization.

Change: The mentioned novelties are highlighted in the Chapter 1, Introduction, of the revised paper, as well in Chapter 5, conclusions.

Please see in Chapter 1: Introduction: Lines 49-55 and Chapter 5: Conclusions, Lines: 189-197 and lines: 199-200.

  1. The research aims to minimize the total cost of energy consumption and peak load in collective residential buildings, but it does not show the percentage of energy savings or peak load reduction obtained from implementing the model proposed in the case under analysis. It would be appropriate to compare the results obtained in the research with the results of other studies of the state of the art, in order to be able to evaluate the potential of the proposed model with respect to other solutions and to highlight its advantages and limitations.

Change: In comply with this comment, Table 3 was added in which the percentage of total cost and peak load reduction are comprised.

Please see Chapter 4: Case study, Table 3 and the text in Lines:181 to 187.

  1. It would be appropriate to compare the results obtained in the research with the results of other studies of the state of the art, in order to be able to evaluate the potential of the proposed model with respect to other solutions and to highlight its advantages and limitations. For example, estimating the arrival time of electric vehicles is difficult, as atmospheric conditions can influence how often a vehicle is used on the road.

Reply: Thank you for this comment. In fact, the estimation of arrival times and the consideration of the effect of atmospheric conditions are beyond the scope of this proposed paper. Authors aim to consider this comment as subject of a new research work.

  1. In addition, the model considers that all users have an electric vehicle, which is a difficult limitation to achieve at present.

Reply: At the presented model, to say that one user has not an EV, we can set the capacity of the corresponding EV to zero.
Change: This point is highlighted in the new version of manuscript. Please see Chapter 3: Problem Description and Mathematical Model, Lines: 128 - 129.

  1. Besides, the abstract does not present numerical results obtained during the research.

Change: Thank you for your suggestion. The Abstract is revised accordingly.

Please see Abstract: Line 11 to 13.

  1. On the one hand, in this type of study it is convenient to analyze the profitability of the proposed installation compared to other solutions already carried out.

Reply: This point was carried out via Table 3, Chapter 4: Case Study.

  1. On the other hand, in the proposed case, the characteristics of the photovoltaic installation and the solar radiation values are not presented.

Reply: Authors have used a real data base concerning PV generation in Porto, Portugal, where each PV generation has 3.68 kW.

Change: Despites not having the characterization of photovoltaic installation and the solar radiation values, it was emphasized that authors have used real data concerning load consumption and PV generation.

 Please see Chapter 4: Case Study, Line: 140.

Reviewer 3 Report

The article is not well organized.

What is the architecture of this system (energy sources, inverters, storage elements etc.)?

Some information is missing in this article: 1) What is the total power produced by PV (case studied in this article)? 2) what is the storage capacity in this study?

 The energy management method must be compared with other methods in order to show the efficiency of this technique.  

How did the authors calculate SOC in this study? what is the capacity for injecting energy into the network in the V2G framework or V2B in this study?

Simulation results are missing.

 References are poor.

The authors can use the references below to improve their article.

  • A Smart Cyber Physical Multi-Source Energy System for an Electric Vehicle Prototype, ELSEVIER 2020
  • Grid of Hybrid AC/DC Microgrids: A New Paradigm for Smart City of Tomorrow, 2020 IEEE 15th International Conference of System of Systems Engineering (SoSE).

Author Response

The authors would like to express their gratitude to the reviewer 3 for his valuable comments that give us the opportunity to improve the paper. The authors present their response below.

1. What is the architecture of this system (energy sources, inverters, storage elements etc.)?

Reply: This work considered real data base concerning low voltage customers consumption and PV generation. In addition it was considered the intensive use of EVs and a central storage system. All the information about the architecture of the energy resources (e.g. inverters) are out of the scope of the proposed paper.

Change: Related this topic some changes were made in equations (8) and (9), as well as in Line 137.

2. Some information is missing in this article: 

2.1) What is the total power produced by PV (case studied in this article)?

Change: Thank you for your question. This missing information was added in Chapter 4: Case Study, line 137.

 2.2) what is the storage capacity in this study?

Change: Authors have highlighted the information about capacity storage value in equation (9), of the Chapter 4.

3. The energy management method must be compared with other methods in order to show the efficiency of this technique. 

Change: The Authors are grateful for reviewer comment. In comply with this comment, Table 3 was added in which the percentage of total cost and peak load reduction are comprised.

Please see Table 3 and the text in Line 181 to 187.

4. How did the authors calculate SOC in this study? What is the capacity for injecting energy into the network in the V2G framework or V2B in this study?

Reply: The initial EV SoC is randomly as mention in Line 149-150 and by eq. (7b) the SoC is calculated for each time-slot. The maximum capacity of the EV discharge is considered in Eq. 7e.

5. Simulation results are missing.

Change: Thank you for your valuable comment. In Chapter 4, Table 3 was added in which the percentage of total cost and peak load reduction are comprised. Please see Table 3 and the text in Line 181 to 187.

6. References are poor.

The authors can use the references below to improve their article.

  • A Smart Cyber Physical Multi-Source Energy System for an Electric Vehicle Prototype, ELSEVIER 2020
  • Grid of Hybrid AC/DC Microgrids: A New Paradigm for Smart City of Tomorrow, 2020 IEEE 15th International Conference of System of Systems Engineering (SoSE).

Change: Thank you very much for your suggestion. The suggested papers were taken into consideration (References [1] and [7]).

Round 2

Reviewer 1 Report

Manuscript after revision is ready to publish it. 

Author Response

The authors would like to express again their gratitude to the reviewer 1 for his/her valuable comments. 

Reviewer 2 Report

The results of the article have been compared with a similar model proposed by the same authors. It is not clear whether the improvement in the results originated from the proposed model or because in the mentioned reference study they did not consider the flexibility of the power contracted for each flat or the process of unloading the EVs. This model should be compared with others used to optimize energy management in Smart Buildings, in order to determine if the model proposed in the article allows for the improvement of those formulated in the existing literature.

In addition, it would be advisable to discuss, based on the results, the advantages of Multi-Objective Mixed-Binary Linear Programming compared to Mixed-integer Linear Programming. This would be necessary to determine the novelty of the research.

Author Response

The authors would like to express their gratitude to reviewer 2 for his/her valuable comments. The authors have presented their responses below.

  1. The results of the article have been compared with a similar model proposed by the same authors. It is not clear whether the improvement in the results originated from the proposed model or because in the mentioned reference study they did not consider the flexibility of the power contracted for each flat or the process of unloading the EVs. This model should be compared with others used to optimize energy management in Smart Buildings, in order to determine if the model proposed in the article allows for the improvement of those formulated in the existing literature.

Reply: Thank you for your comment. The results of scenario 1 of our last paper (Ref. [12]) were considered as a ground to assess our model in this paper. Moreover, the following ideas in this paper are not considered in scenario 1 of [12]:

  • Charge and discharge EVs and schedule the charge/discharge times to move demand from low-demand times to high-demand times and BESS (Battery Energy Storage System) usage.
  • Enable flexibility of the contracted power, i.e., considering only one contract power for the whole residential building in order to reduce the electricity costs.

We mention that the reported improvement in the results was obtained because of both of the above considerations.

Let us add that, the above characteristics of the proposed method lead to difficulty in making a fair comparison with other works in smart buildings. Also, please note that our main point in this paper is to study the problem in a multi-objective framework. In multi-objective frameworks, the aim is not to provide a better solution in terms of one objective function. As a result, make a comparison between a multi-objective model and a single objective model is not sensible. The results of a multi-objective model are provided for decision-makers to select a Pareto solution.

In addition, to make a comparison with other existing models in the literature, it is required to have the same, or very similar, case study. That is why authors have used the case study from work [12].

Change: In order to consider the reviewer concern, the following paragraph is added to the revised paper:
Finally, in order to show the efficiency of the proposed model, the obtained results (total consumption costs and the maximum value of the demanded load) were compared with the reference case study that neither considered the flexibility of contracted power for each apartment nor the EVs discharging process [12].”
Please check at: Line 185 – 188.

  1. In addition, it would be advisable to discuss, based on the results, the advantages of Multi-Objective Mixed-Binary Linear Programming compared to Mixed-integer Linear Programming. This would be necessary to determine the novelty of the research.

Reply: Thank you for your suggestion. Generally, the multi-objective framework provides more suitable results that can be considered by decision-makers. Indeed, decision-makers, based on the existing resource, select a solution among the obtained Pareto points that is a tread-off between cost and peak load demand.

Please note that the multi-objective problem provides a set of solutions that is an approximation of the entire Pareto front. The solutions of single-objective problems belong to the reported Pareto point of the multi-objective problem. For example, when epsilon=0, the objective function is minimizing the Total Costs and by considering epsilon=1, the Peak Loads is minimizing. Consequently, the solutions of single objective problems are compared with other solutions by decision-maker. Accordingly, the multi-objective framework leads to more robustness in decision making.

Change: Concerning this comment, for more clarifying, the following paragraph was added to the new version of the paper:

Note that, when “epsilon”=0, the single-objective problem is solved in order to minimize the total energy cost (the optimal achieved value is 39.0317 Euros). when “epsilon”=1, the single-objective problem is solved in order to minimize the demanded peak load (the optimal achieved value is 4.7033 kWh).”

Please check at: Line 157 – 159.

Reviewer 3 Report

The authors did not use the reference n°7 cited in their article properly. This is a smart management system in an electric vehicle and not in a smart building. You have to look at all the references carefully.

Here is your text: So far, extensive research work has been done on the topic of building energy management in order to schedule renewable energy sources with different goals [2–7].

Please correct it!

Author Response

The authors would like to express their gratitude to the reviewer 3 for his valuable comments. The authors present their response below.

  1. The authors did not use the reference no7 cited in their article properly. This is a smart management system in an electric vehicle and not in a smart building. You have to look at all the references carefully. Here is your text: So far, extensive research work has been done on the topic of building energy management in order to schedule renewable energy sources with different goals [2–7]. Please correct it!

Reply: The Ref. [7] suggested by the reviewer is not entirely relevant. However, the suggested reference was added to the manuscript during the first revision. We admit that the place of this reference was not proper. Accordingly, in the new version of the paper, we cite Ref. [7] in another place, and a relevant discussion is added.

Change: The comment is considered in the new version of the paper by adding the following sentence:
“…as well as other studies largely focused on the energy resource, such as the electrical vehicle [7].”
Please check at: Line 24 – 25.